# Development of an AI-Empowered Novel Digital Monitoring System for Inhalation Flow Profiles [note 1]

**DOI:** 10.3390/s25144402

**Published:** 2025-07-15

**Authors:** Ziyi Fan, Yuqing Ye, Jiale Chen, Ying Ma, Jesse Zhu

**Affiliations:** 1Department of Biomedical Engineering, University of Western Ontario, London, ON N6A 3K7, Canada; zziyifan@uwo.ca (Z.F.); yma9@uwo.ca (Y.M.); 2Department of Chemical Engineering, Nottingham Ningbo China Beacons of Excellence Research and Innovation Institute, The University of Nottingham Ningbo China, Ningbo 315100, China; jiale.chen@nottingham.edu.cn; 3Suzhou Inhal Pharma Co., Ltd., Suzhou 215125, China; 4School of Chemical and Biomolecular Engineering, Eastern Institute of Technology, Ningbo 315200, China

**Keywords:** dry powder inhalers, digital monitoring system, machine learning, flowrate estimation, inhalation flow profile

## Abstract

The use of dry powder inhalers (DPIs) represents a cornerstone in the treatment of chronic pulmonary diseases. However, suboptimal inhalation techniques, including inadequate airflow rates, have been a persistent concern for achieving effective therapeutic outcomes, as many patients remain unaware of their insufficient inhalation performance. As an effective strategy, a digital monitoring system, coupled with dry powder inhalers (DPIs), has emerged to estimate flow profiles and provide inhalation information. The estimation could be further facilitated by the application of artificial intelligence (AI). In this work, a novel digital system to primarily monitor pressure during DPI usage was successfully designed, and advanced machine learning (ML) techniques were then employed to estimate inhalation flow profiles based on the captured data. Four optimal machine learning models were selected for subsequent inhalation parameter prediction, given their superior generalization ability. By using these models, inhalation flow profiles could be successfully estimated, with an excellent accuracy of 97.7% for Peak Inspiratory Flow Rate (PIFR) and 95.2% for inspiratory capacity (IC). In summary, the pressure-based digital monitoring system empowered by AI techniques could be successfully applied to assess inhalation flow profiles with excellent accuracy.

## 1. Introduction

Dry powder inhalation has been one of the most commonly employed methods in clinical practice, especially for the treatment of pulmonary diseases, such as asthma and chronic obstructive pulmonary disease (COPD), which affect over half a billion patients worldwide [1]. Most dry powder inhalers (DPIs) available on the market are breath-activated, highly relying on inspiratory airflow to fluidize and disperse powdered formulation [2]. To achieve optimal therapeutic outcomes, the dry powder inhalation system requires patient adherence to prescribed dosing regimens, correct inhaler usage, and adequate inspiratory airflow rate [3]. Despite decades of efforts aimed at improving patient adherence and inhaler techniques, a substantial proportion of patients fail to follow prescribed dosing regimens, with misuse reported in up to 80% of cases [4,5].

To enhance disease management and optimize treatment outcomes, digital monitoring systems that integrate sensor technologies and mobile connectivity [6] have been developed. These systems could continuously record inhalation process and extract critical inhalation parameters, providing real-time feedback to patients regarding inhaler usage [3,7,8]. Such day-to-day feedback could improve patient adherence and inhaler techniques by making individuals more aware of suboptimal inhalation patterns, such as insufficient inspiratory flow that could impair drug delivery efficiency [9,10,11]. Additionally, the digital systems allow for remote sharing of inhalation data with healthcare providers, facilitating timely adjustments to treatment strategies and, therefore, achieving better control of diseases. Overall, the digital monitoring system with active feedback presents an opportunity to improve the therapeutic efficacy of inhaled therapy, reduce the exacerbations of diseases, and lower the frequency of hospital emergency admittance [12,13].

Among the issues associated with inhaler misuse, the inability of patients to achieve a sufficient airflow rate is a critical challenge that impairs effective pulmonary drug delivery. To indicate a patient’s capability to achieve adequate inhalation, peak inspiratory flow rate (PIFR), defined as the maximal flow rate during an inspiratory maneuver, is frequently used as a crucial parameter of inhalation [14,15]. A PIFR above 30 L/min, ideally above 60 L/min, is recommended to ensure effective powder dispersion and dose delivery [3,16]. On the contrary, an insufficient PIFR may lead to suboptimal drug deposition and compromised therapeutic effects. In addition to PIFR, inspiratory capacity (IC) has also been used as a key parameter to indicate the inhalation of patients. A reduced IC is often associated with respiratory exacerbations, as it reflects limitations in lung volume and capacity, which may worsen during acute episodes of airway obstruction or disease progression [17].

Given the critical role of sufficient airflow rate in effective pulmonary drug delivery, studies have investigated methods for accurately estimating inhalation airflow rates. Among these methods, two of the most prominent approaches involve audio-based and pressure-based measurements. Audio-based flow rate estimation relies on microphones to capture sound signals related to inhalation. This method has been applied to remotely monitor the inhalation techniques of inhaler usage [3,8,18,19]. For instance, in the work by Holmes et al., audio signals were recorded using the Inhaler Compliance Assessment (INCA^TM^) device attached to the Ellipta™ DPI [3]. The acoustic envelope (how a sound’s amplitude changes over time) of each inhalation was extracted, and regression models were applied to determine the best relationship between the inhalation audio signal and the corresponding flow rate. However, the audio-based method is highly susceptible to ambient noise, which will interfere with signal quality and lead to inaccurate estimation of the flow profile [20]. Particularly, at low inhalation airflow rates, flow signals can be too weak to distinguish from ambient background noise, making accurate prediction even more difficult [10,21]. The method also requires advanced yet complex and intricate signal processing techniques to effectively filter noise and extract features.

The pressure-based method represents another feasible strategy, typically employing pressure sensors to measure differential pressure signals across the inhaler to estimate flow rate. One example of commercially available digital systems is Digihaler^®^ (Teva Pharmaceutical, Parsippany, NJ, USA), which incorporates a pressure sensor inside the inhaler to estimate inhalation flow rate. Another representative example is the Sensirion^®^ (Sensirion AG, Stäfa, Switzerland) clip-on prototype, which utilizes a differential pressure sensor to record signals associated with inhalation and airflow rate. Although the functional capabilities of these systems have been described, limited research has been published on the methodologies for estimating inhalation flow rates using such pressure-based measurements.

In recent years, artificial intelligence (AI), as a powerful cutting-edge technology, has been applied in inhaled therapy due to its prominent advantages of recognition, prediction, and data analysis [22]. Machine learning (ML), as a pivotal AI technique, has been utilized to establish correlations between multidimensional sensor data (such as sound signal, pressure signal) and inhalation parameters (such as PIFR and IC) [23,24,25]. For example, FAKOTAKIS et al. revisited sound pattern recognition with machine learning techniques (decision tree model, hidden Markov model, and random forest), and the results showed excellent classification efficiency on inhalation-related activities, although it did not exhibit that one model clearly outperformed the others [20]. Alam et al. also applied machine learning models to predict forced expiratory volume in 1 s (FEV1, a key indicator of lung function) with an accuracy of 85%, by recording the voice signal of asthma patients, although not indicated for DPI use monitoring [26].

Based on our review of current digital monitoring systems for dry powder inhalers (DPIs), it was found that there are proposed AI-empowered systems that employ an acoustic method and use AI techniques to categorize inhalation activities in order to address misuse issues [27,28]. However, to date, there is no published reporting on the integration of AI techniques with a pressure-based system for inhalation flow estimation. Therefore, a novel digital system that utilizes pressure signals was developed, taking advantage of its superior ambient noise resistance. Meanwhile, advanced ML techniques were also integrated to refine the signal processing, enhancing the accuracy and reliability of the system. This proposed system features a compact MEMS sensor that simultaneously measures pressure, temperature, and humidity signals, thereby also enabling multidimensional compensation. The approach not only overcomes the limitations of traditional analysis methods such as linear regression but also highlights the potential of machine learning techniques in improving the estimation of PIFR and IC. This study underscores the significance of integrating AI to enhance the performance of pressure-based monitoring systems in DPIs.

## 2. Materials and Methods

To monitor the inhalation activities, a novel digital monitoring system was developed. Its main part is a custom-designed digital module comprising mechanical frames and electronic components, attached to a capsule-based inhaler, Breezhaler^®^. The digital module, together with the DPI, constitutes a complete digital monitoring system designed to collect inhalation data of patients.

### 2.1. Digital Module Design

Figure 1 presents the trimetric view (left) and the exploded view (right) of 3D models of the DPI digital monitoring system. The 3D models of both the inhaler and the digital module were created by SolidWorks 2023. The digital module includes a base, a case, and a self-designed printed circuit board (PCB). The mechanical frames of the digital module were prototyped using a Stereolithography 3D printer (iSLA660, ZRapid Tech, Suzhou, China) and feature a specially designed pressure detection configuration. The frame design of the digital module ensures its seamless integration with the inhaler while preserving functional integrity of the DPI.

Figure 2 shows a cross-sectional view of the customized digital module with the DPI, Breezhaler^®^ (Novartis, Basel, Switzerland). In the digital module, the case holds a MEMS (Micro-Electro-Mechanical Systems) sensor encased inside a “sensor cell”. During inhalation, ambient air is entrained into the system through the gas inlets of the inhaler. The sensor cell is connected to the airflow pathway, enabling precise pressure measurements for accurate detection of the inhalation flow profile without direct airflow passing through it. At the same time, the airflow stream travels into the circular chamber of the DPI, rotating the capsule and aerosolizing the formulation powders inside. This configuration for pressure detection is specially designed to minimize any interference with the original flow field of the inhaler, preserving the functional performance of the DPI.

The mechanical frames of the digital module house a PCB that integrates a MEMS sensor, Bluetooth module, USB port, battery storage, and an LED indicator. The MEMS sensor captures real-time pressure, humidity, and temperature data, enabling comprehensive monitoring of both user activity and environmental conditions during inhalation. The Bluetooth module, which also functions as a microcontroller unit, processes the collected data and wirelessly transmits it to external devices for further analysis. The battery powers the module, while the USB port supports recharging. The LED indicator serves as a user interface, signaling system readiness and operation status. This compact, integrated design enables efficient and real-time inhalation monitoring.

### 2.2. Signal Collection

The experimental setup for the collection of inhalation data is shown in Figure 3. Pressure signals were directly obtained from the custom-designed digital monitoring system, while the real-time flow rate signals were simultaneously recorded by using a digital bidirectional flowmeter (model SFM3000, Sensirion AG, Stäfa, Switzerland) with a 14-bit resolution and a flow range of ±200 L/min. The flowmeter was interfaced with the same data acquisition laptop, enabling real-time visualization of flow rates (served as referenced flowrate values) through the SENSIRION^®^ ControlCenter 1.40.2 DataViewer software. Both the pressure sensor and the flow sensor were configured at a sampling rate of 20 Hz, based on the previous work by Tayler et al., which indicates that a cut-off frequency of 4 Hz is sufficient to capture the rapid change in signals during inhalation [10].

Eight healthy volunteers participated in this study with informed consent. To capture natural inhalation behaviors, no specific instructions on inhaler usage techniques were provided. Each participant performed 20 inhalations, resulting in a total of 160 inhalation recordings. Each recording consisted of multivariate time-series data, yielding approximately 328 data points (82 timestamps × 4 channels) per recording. The final dataset comprised 13,168 rows in tabular format, including synchronized measurements of flow rate, pressure, temperature, and humidity, along with corresponding timestamps.

### 2.3. Data Preprocessing and Model Selection

Typically, signal preprocessing is performed to standardize the input for machine learning models. In this study, the raw signals acquired from the digital monitoring system, including pressure, temperature, humidity, and timestamp, were intentionally subjected to minimal preprocessing. This approach was adopted to allow the models to learn underlying patterns and perform inherent denoising autonomously during training. Specifically, the pressure and flow signals were only processed using a second-order low-pass Butterworth filter with a 4 Hz cut-off frequency to enhance estimation accuracy, as recommended by Taylor et al. [10].

Moving on to the model selection, existing ML algorithms, including individual base learners and advanced heterogeneous ensembles, were selected, applied, and optimized for the subsequent prediction of inhalation airflow. The base learners are individual machine learning models that function as the building blocks for ensemble learning models, while heterogeneous ensembles are a type of ensemble learning that combines predictions from multiple different types of base learners [29,30]. A comparative analysis of eight base learner algorithms was conducted as a foundational approach to understand and predict inhalation flow profiles. Following the evaluation of these individual algorithms, advanced ensemble learning techniques were subsequently assessed to explore their potential improvements in predictive accuracy and robustness.

ML models were trained and evaluated using a 5-fold cross-validation approach, with the exception of the blending ensembles, which employed a holdout set strategy for evaluation. To implement the cross-validation, the collected dataset was divided into a training set and a testing set, with 75% of the data allocated for training and the remaining 25% for testing. Moreover, ML model hyperparameters were tuned using a grid search approach, enabling systematic exploration and optimization of model parameters to achieve robust performance [31].

All model training and evaluation were performed on a personal computing platform equipped with an AMD Ryzen 7 4700U CPU @2.00 GHz processor without GPU acceleration. The machine learning algorithms were implemented primarily using the scikit-learn library (Python 3.9). Due to variations in model complexity and scope of grid search parameter tuning, the total training time differed across algorithms. A detailed summary of the training durations for each model is presented in Appendix A.

#### 2.3.1. Base Learner Algorithms

In this section, eight base learners were examined, each tested with diverse parameters. The Decision Tree algorithm, due to its simplicity and interpretability, was used for subsequent comparisons [32]. Support Vector Machine (SVM), which is renowned for its ability to maximize the margin between classified data points, was employed to model nonlinear relationships within the dataset and enhance model performance in complex scenarios [33]. Gaussian Process Regressor (GPR) was used to generate probabilistic predictions along with uncertainty estimates. This capability is particularly valuable when dealing with small-sized samples, as it aids in understanding the variability that is inherent in inhalation flow profiles.

To improve the accuracy and robustness of predictive models, various homogeneous ensemble learning techniques were applied. Homogeneous ensembles aggregate multiple instances of the same type of base learner. They are expected to reduce overall error and enhance predictive accuracy compared to individual base learner models. For example, bagging-based algorithms, such as Random Forest (RF) and Extra Trees Regressor (ETR), were implemented to reduce variance and stabilize predictions. Boosting-based algorithms, including AdaBoost, XGBoost, and GradientBoosting algorithms, were utilized to sequentially correct errors and strengthen overall model performance. In this study, the author grouped the individual Decision Tree, SVM, GPR, with the homogeneous ensemble models (RF, ETR, AdaBoost, XGBoost, and GradientBoosting) together as base learner models to preliminarily evaluate their performance.

#### 2.3.2. Heterogeneous Ensemble Algorithms

Despite the trials in Section 2.3.1, homogeneous ensembles may not fully capture the diversity of insights offered by distinct modeling approaches. To overcome this limitation, heterogeneous ensemble algorithms were then introduced, combining different types of base learners to harness their complementary strengths, thus further improving predictive performance and robustness. Based on the combination strategies, heterogeneous ensemble models can be further categorized into voting ensembles, stacking ensembles, and blending ensembles.

Voting is one of the fundamental heterogeneous methods in machine learning. In the training stage, as shown in Appendix A, multiple base learners are trained independently on the same dataset. Following this, the final prediction in the output stage is commonly obtained by calculating the average predictions generated by each base model in the training stage. This averaging approach helps to mitigate individual model errors as well as enhance the robustness and generalizability of the ensemble’s prediction.

The framework for the stacking ensemble strategy is illustrated in Appendix A. Stacking is an advanced ensemble technique by which multiple base models are trained, and their predictions are then integrated through a meta-model to enhance predictive accuracy and generalizability. In this study, a 5-fold cross-validation strategy was employed to train the meta-model, therefore improving its robustness and mitigating overfitting. More details of this structural approach are available in the literature [34].

Blending ensemble shares similarities with stacking, as both methods use base learners to generate predictions, which are then treated as new features for a meta-model that makes the final prediction. However, blending differs by incorporating a holdout set (validation set) to train the meta-model on these new features. In this study, the holdout set was created by splitting the original training data into an 80:20 ratio. The blending framework is illustrated in Appendix A.

### 2.4. Model Evaluation

In this study, model performance across various machine learning models was evaluated using four key metrics: coefficient of determination ($R^{2}$), mean absolute error (MAE), mean squared error (MSE), and root mean squared error (RMSE). The metrics are defined by the following Equations (1)–(4):(1)$$R^{2}=1-\frac{\sum {\left(\hat{y_{i}}-y_{i}\right)}^{2}}{\sum {\left(\bar{y}-y_{i}\right)}^{2}}$$(2)$$MAE=\frac{1}{N}\sum_{i=1}^{N}\left|\hat{y_{i}}-y_{i}\right|$$(3)$$MSE=\frac{1}{N}\sum_{i=1}^{N}{\left(\hat{y_{i}}-y_{i}\right)}^{2}$$(4)$$RMSE=\sqrt{\frac{1}{N}\sum_{i=1}^{N}{\left(\hat{y_{i}}-y_{i}\right)}^{2}}$$
where $\hat{y_{i}}$ denotes the predicted value, $y_{i}$ is the actual (ground truth) value, $\bar{y}$ is the mean of all actual values, and N is the total number of samples.

### 2.5. Estimation and Evaluation of Inhalation Parameters

Among the evaluated ML models, the best models were selected to plot the airflow rate profiles of inhalation using the collected data. To assess the estimation accuracy on the flow profiles, two key parameters, PIFR and IC, were used. The two flow parameters were obtained from the flow profiles, where the PIFR is the peak flow point of the inhalation flow curve, and the IC is the area under the inhalation flow profile curve that corresponds to the total inhaled air volume. This IC was obtained through trapezoidal numerical integration of flow profiles. The accuracy of the parameter estimation was determined by comparing the estimated PIFR and IC with their corresponding reference values for each inhalation. The calculation of estimation accuracy for PIFR and IC is detailed in Equations (5) and (6):(5)$${PIFR}_{Accuracy}\left(\%\right)=100-\frac{\left|{PIFR}_{est}-{PIFR}_{act}\right|}{{PIFR}_{act}}\times 100$$(6)$${IC}_{Accuracy}\left(\%\right)=100-\frac{\left|{IC}_{est}-{IC}_{act}\right|}{{IC}_{act}}\times 100$$

The results were shown as the mean with standard deviation (mean ± sd). For model comparisons, the Wilcoxon signed-rank test was employed to assess differences in prediction accuracy, with *p* < 0.05 considered statistically significant.

## 3. Results

This section presents the results of a series of machine learning models developed to predict inhalation flow profiles from sensor data. A comprehensive set of base learner algorithms and advanced ensemble algorithms was implemented to assess their effectiveness in capturing the underlying inhalation dynamics. The models were evaluated using multiple performance metrics, and the best-performing models were selected for further predictions.

Firstly, data recorded by the digital monitoring system were collected and are presented in Figure 4. Figure 4a illustrates the scatter plots depicting the relationships between reference flowrate and four types of signals (pressure, temperature, humidity, and time) collected. The negative values of the reference flowrates only suggest direction defined by the SENSIRION^®^ flowmeter. It is evident that the correlations between referenced flowrate and collected signals do not follow a simple linear pattern. This nonlinearity may arise from complex fluid mechanics, individual variations in inhalation behaviors, sensor response characteristics, as well as the influence of ambient environmental conditions. Consequently, AI-based approaches were adopted as they are better suited to modelling such nonlinear and multifactorial relationships, enabling accurate prediction of real-world inhalation flow rates.

Figure 4b presents a case sample of collected pressure and reference airflow rate profiles. The reference airflow rates were used not only for model training but also for evaluating the accuracy of the prediction. Figure 4c shows the overall flow of AI techniques for prediction. With the collected data, pressure, temperature, humidity, and time, as input variables, basic and advanced machine learning algorithms were developed and used for prediction. The timestamp was also incorporated as one of the inputs due to the intrinsic time-series nature of inhalation airflows. The feature is important for identifying the onset and termination of inhalation events.

### 3.1. Evaluation of Machine Learning Models

#### 3.1.1. Using Base Learners

Following data collection, the machine learning models were developed and evaluated. In this section, eight base machine learning (ML) algorithms were employed to model the relationship between a dependent target variable (flow signal) and a set of independent features (pressure signal, temperature, humidity, and timestamp). Prior to model evaluation and prediction, ML algorithms need to be optimized by fine-tuning the corresponding hyperparameter set, defined as a configurable setting that governs the learning process and how models learn [35]. The optimal hyperparameters, as shown in Appendix A, were obtained based on the highest R^2^ score that was achieved using grid search.

Moving on to the model evaluation, it is to assess how well a trained model performs. To ensure the accuracy and generalizability of models, the evaluation was performed on three distinct parts: cross-validation, testing, and training sets, and the evaluation results are summarized in Table 1. The model evaluation primarily focused on performance metrics obtained from cross-validation and testing datasets. With respect to the evaluation metrics of model performance, an R^2^ value approaching 1 suggests a strong model fit, while lower values in error metrics (MAE, RMSE, MSE) indicate superior accuracy in the model’s predictability.

In the evaluation of the cross-validation sets, both RF and AdaBoost stood out as top performers and reached the same highest R^2^ of 0.944, indicative of excellent predictive accuracy and consistency across different data subsets. In contrast, GradientBoosting exhibited the lowest R^2^ performance at 0.898, suggesting less effective generalization compared to other models. In terms of error metrics, the RF demonstrated superior generalization capabilities, evidenced by its lowest error values (MAE 2.788, RMSE 6.383, MSE 40.781). Conversely, the GradientBoosting algorithms recorded relatively large errors and exhibited significantly larger deviations, reflecting greater variability in their performance across different folds.

Upon analyzing the evaluation results on the testing sets, all the algorithms achieved R^2^ values above 0.91, with the highest R^2^ of 0.955 for the AdaBoost model, followed by 0.952 for RF and 0.951 for ETR. In terms of MAE, the results varied across models, with the RF model displaying the lowest MAE of 2.448, indicative of minimal average deviations, whereas GradientBoosting exhibited the highest MAE of 3.803, reflecting the largest average absolute errors on testing sets. The RMSE metrics also showed variability among the models, ranging from 5.635 for AdaBoost, which denoted highly accurate predictions with minimal error spread, to 7.629 for the SVM, which suggests a broader spread of errors on the test set. The MSE extended from 31.759 for AdaBoost to 58.207 for SVM, highlighting the squared average deviations of the predictions.

To clearly indicate the overall performance of these models, Figure 5 presents the bar plots and radar plots of the evaluation results. Figure 5a demonstrates the results of error metrics of the models with standard deviations, and one can see that the Random Forest (RF) model exhibited the highest R^2^ values as well as the lowest error metrics values with minimal variation in errors. The radar plot shown in Figure 5b visualizes the performance metrics (RMSE Mean, MSE Mean, MAE Mean, and R^2^) for each model, normalized using the Min-Max scaler for direct comparability. The results indicated that the RF model, with points furthest from the origin, achieved the highest performance across these metrics. In contrast, the Gradient Boosting model is closer to the origin, demonstrating the lowest performance among the models evaluated. The RF model also stood out among the results of the testing sets, with the lowest MAE, and showed top performances regarding other scores. Therefore, Random Forest was selected as the best model for predicting flowrate and for further analysis regarding inhalation parameters.

#### 3.1.2. Using Heterogeneous Ensemble Models

In this section, advanced heterogeneous ensemble models were constructed using the four best-performing base learner models. Based on the results presented in Section 3.1.1., the top four models demonstrate higher R^2^ values. Random Forest, AdaBoost, Extra Trees Regressor, and XGBoost were identified as optimal candidates for inclusion as base models in the construction of heterogeneous ensemble frameworks, instead of only selecting the best performer (RF model). The structures of the heterogeneous ensemble frameworks are detailed in Appendix A.

Appendix A provides additional insights into the prediction performance of these four models. The analysis of the prediction plots on the testing sets indicates that all four models demonstrated high prediction accuracy in the high flowrate range (≥70 L/min), moderate accuracy in the medium range (30–70 L/min), and noticeably reduced performance in the low flowrate range (≤30 L/min). The performance in the low flowrate range also aligns with trends reported in a previous study [36].

Similar to the evaluation on base learner models in Section 3.1.1., the evaluation on the advanced heterogeneous ensemble algorithms also focused on the cross-validation and testing datasets. Table 2 presents the evaluation metrics of cross-validation sets for these heterogeneous ensemble models. Collectively, no significant increase or decrease in these error metrics for the heterogeneous models was observed when compared with the top four base models. Voting Ensembles 1 and 2, as well as Stacking Ensembles 1 and 3, are relatively top performers among the developed ensembles, showing slightly higher R^2^. It is important to note that, due to the unique structure of the blending ensemble method described previously, its performance was evaluated only on the out-of-sample testing set, rather than through cross-validation.

Table 3 shows the performance of heterogeneous ensemble models, including voting ensemble, stacking ensemble, and blending ensemble, evaluated on the testing sets. Among the 14 heterogeneous ensemble models, Stacking Ensemble 3 (RF + AdaBoost + XGBoost; Meta model = ETR) demonstrated the highest R^2^ and the lowest values of error metrics among the testing cohorts, indicating its outstanding performance. This model also maintained consistently strong performance across cross-validation sets, substantiating its predictive reliability and generalizability. Of the four voting ensemble models, Voting Ensembles 1 and 2 performed slightly better, with Ensemble 1 marginally outperforming Ensemble 2, showing comparable prediction performance. As for blending ensemble models, model 3 illustrated the best performance based on R^2^ and other error metrics.

Using those heterogeneous models, the scatter plots depicting the estimated and referenced airflow rates were also generated, as shown in Appendix A. The scatter plots also emphasized the prediction accuracy of Voting Ensemble 1, Stacking Ensemble 3, as well as Blending Ensemble 3, although it is visibly difficult to differentiate the scatterability of these spots.

Overall, the comprehensive evaluation of the results for both cross-validation and testing sets indicates that Stacking Ensemble 3 emerged as the highest-performing model among all of the heterogeneous ensemble models. Voting Ensemble 1 and Blending Ensemble 3 demonstrated strong performance within their categories. These superior models will be employed for further prediction of airflow profiles and key parameters.

### 3.2. Prediction and Evaluation of Inhalation Parameters

In this section, the top-performing base learner (RF) and the top-performing heterogeneous ensembles (Voting Ensemble 1, Stacking Ensemble 3, and Blending Ensemble 3) were used to predict the inhalation flow profiles, respectively.

Table 4 shows the estimation accuracy of PIFR and IC by using the top four models. The Random Forest model achieved the highest accuracy for both PIFR (97.7 ± 2.9%) and IC (95.2 ± 9.0%), indicating strong agreement between predicted and actual values. The other ensemble models yielded PIFR accuracies between 95% and 97% and IC accuracies between 94% and 95%. Statistical analysis revealed significant differences (*p* < 0.05) between Random Forest and the three ensembles, while no significant difference was found between Random Forest and Stacking (*p* = 0.68) for IC prediction. These results suggest that ensemble methods provided no advantage over the best-performing single learner, with Random Forest consistently outperforming all models.

Additionally, the error margin for PIFR ranged from 2.9% to 5.1%. Considering that device-specific variability and intra-patient variability in PIFR measurements are typically within 10% [37,38,39], the prediction error of current models is deemed clinically acceptable. The prediction error for IC was slightly higher, with a standard deviation ranging from 9.0% to 9.5%. Although this value remains below the 10% margin, there is currently no established clinical reference to confirm whether this level of error is acceptable.

To further explore the clinical relevance of model predictions, a simple binary classification analysis was performed using a clinically meaningful threshold of PIFR ≥ 50 L/min for Breezhaler^®^ [40]. This analysis illustrates how small prediction errors can affect the determination of sufficient inhalation effort and supports the potential of the model to inform clinical decision-making (see Appendix A). Since PIFR is widely recognized as the most clinically relevant parameter, this analysis was limited to PIFR and not extended to IC.

Figure 6 shows a case example of an estimated airflow rate profile using these optimal models and corresponding PIFR and IC (predicted), with relative error results shown in Appendix A. The case inhalation sample was selected among the recordings at random. Its collected pressure data profile is shown in red, and its simultaneously collected airflow profile (as a reference) is shown in green (Figure 6). Based on the collected pressure data, the selected best models were applied to estimate the inhalation flow profiles, which are shown in different line colors in Figure 6. One can see that the predicted flowrate profiles using these optimal models are close to the reference airflow rate profile collected.

Additionally, the ground truth PIFR (reference) was 81.342 L/min at 1.50 s (represented by a red star), and the IC (reference) was 2.110 L shown in Figure 6. In comparison to the referenced PIFR and IC, the predicted PIFR and IC by the four models were close to the references, with relative errors within 3.0%, confirming the capability of the optimized models to generate estimates of inhalation parameters that are closely aligned with the actual values.

## 4. Discussion

This study presents a novel digital monitoring system and also provides a feasible approach to predict inhalation flow profiles by analyzing pressure-based signals. By leveraging various basic and advanced machine learning algorithms, the optimal models could achieve high prediction accuracy by addressing some inherent challenges from (MEMS) sensor.

In this research, the development of an accurate inhalation flow monitoring system requires balancing hardware functions (sensor performance, mechanical design) and software capabilities (algorithms). On the hardware side, the inherent issues of MEMS pressure sensors, such as hysteresis, noise, signal drift, and nonlinearity, pose significant challenges in capturing the dynamics of inhalation. In addition, mechanical design, another aspect of hardware design, also needs to be engineered to minimize interference with the original flow field of the dry powder inhaler (DPI). In this work, the specially designed pressure detection configuration ensures that the inhalation process remains unaltered, allowing for authentic acquisition of pressure signals without affecting inhaler resistance to inspiratory airflow. Despite careful mechanical and sensor design, limitations in hardware performance are inevitable. Therefore, software algorithms play a crucial role in compensating for these hardware imperfections. Signal processing techniques and machine learning algorithms are employed to extract meaningful patterns from noisy, imperfect raw data. By learning complex relationships between pressure signals and flow profiles, these algorithms enable accurate estimation even in the presence of sensor noise and mechanical inconsistencies.

Despite the advantages of algorithms, software-based solutions come with their own challenges, such as computational complexity, real-time processing requirements, and the dependency on high-quality training data to achieve robust performance. Thus, an optimal trade-off involves selecting sensor hardware that provides acceptable baseline accuracy, complemented by advanced software algorithms tailored to mitigate hardware limitations. This hybrid approach, combining thoughtful mechanical design and well-tuned algorithms, ensures the excellent precision required for the inhaler monitoring application. By carefully balancing the strengths and shortcomings of hardware and software, the system can achieve high accuracy without introducing excessive complexity or cost, making it suitable for large-scale deployment.

To make up for the shortness of the MEMS sensor, both basic and advanced machine learning models have been tested to optimize the model and improve the prediction accuracy of the inhalation flow profile. Among the base learner algorithms, bagging-based ensemble models, including Random Forest (RF) and Extra Trees Regressor, were found to outperform other algorithms. These bagging-based models excel in tasks involving complex and nonlinear relationships due to their ability to reduce variance while maintaining robustness [41]. Conversely, boosting-based models exhibited suboptimal performance, with the exception of AdaBoost. This could be attributed to the limited size of the dataset, as boosting-based algorithms generally require larger datasets to achieve optimal training and minimize overfitting [42].

With regard to the advanced heterogeneous ensemble models, they demonstrated potential for improving prediction performance, which is consistent with previous findings on using advanced ensembles [43]. However, their superiority over base learners was not guaranteed, as their performance heavily depended on the structure and design of the ensembles. In this study, stacking ensembles outperformed other heterogeneous methods and the majority of individual base learners. This improvement may stem from the capability of stacking ensembles to effectively combine multiple base learners through meta-models in this task, which not only captures the strengths of each model but also corrects individual model weaknesses. The results emphasize that selecting and positioning base learners is crucial for constructing advanced predictive ensemble models, aligning with findings from previous research [32,44].

When considering the real-world application, the deployment of models should take into account both computational efficiency and hardware constraints. In this study, the RF model not only achieved excellent predictive accuracy but also offers high deployability for real-time applications due to its low inference latency and resource demands. For scenarios requiring more complex models (e.g., voting, stacking, or blending ensembles), a hybrid strategy can be adopted where data acquisition and preprocessing occur on the resource-constrained embedded devices, while inference can be handled by external platforms such as smartphones or cloud servers. This flexible architecture supports low-latency operation, ensures compatibility with low-power hardware, and facilitates scalable integration in both clinical and home-care environments.

One limitation of the present study is the relatively small and homogeneous dataset derived from a limited cohort of healthy participants. Future validation in larger and more diverse populations is necessary to further assess the proposed system and ML models. Ideally, assessment in clinical settings is expected to support the robustness and real-world applicability of the system.

The authors also would like to point out that accurately predicting flowrates in the low range (e.g., <20 L/min) remains a challenge, as evidenced by the scatter plots. The reasons behind the observation may arise from the limited sensitivity of the pressure sensor and the simplistic design of the mechanical frames. These issues can introduce noise and inaccuracies in inhalation flowrate detection. Since the inhalation parameters like PIFR are derived from the predicted flow profile, any inaccuracies in flowrate predictions can propagate to these critical measurements. Despite this limitation, the developed AI-powered digital system should be satisfactory in accurately predicting the PIFR. As noted, a universal threshold applicable across all DPIs is suggested to be 30 L/min, ideally 60 L/min, for effective drug delivery [3,16]. The top-performing models excelled in the prediction of PIFR higher than 30 L/min, which is theoretically sufficient to ensure effective aerosolization and pulmonary drug delivery.

This study highlights the promising application of AI in estimating flow profiles during DPI usage. Despite the promise, advancements are still necessary to further enhance model performance and applicability. A critical next step is to expand the dataset size, as larger and more diverse datasets are essential to improve model accuracy, generalizability, and robustness. The current dataset is limited by the small number of participants with restricted diversity of inhalation signals. Future efforts will focus on collecting data from a broader participant pool. Additionally, applying multi-modal sensor fusion or improving digital module design could further enhance prediction accuracy and reliability.

Another exploration will be advanced deep learning models, such as recurrent neural networks (RNNs) or transformer-based architectures. These advanced models may offer improved performance in capturing complex temporal patterns of inhalation signals. However, it remains a trade-off issue between model complexity and accuracy, as highly complex models may demand greater computational resources and risk overfitting when working with limited datasets. Balancing these factors will be key to developing robust, efficient models that can be deployed in both clinical and real-world environments.

## 5. Conclusions

This study successfully developed a novel digital monitoring system capable of recording pressure, temperature, and humidity signals during inhalation with minimal interference to the functionality of the inhaler. By using advanced machine learning techniques, the digital monitoring system could achieve high-accuracy estimation (>95%) of inhalation flow parameters by integrating and analyzing the captured multidimensional signals.

This work also compares the performance of individual base learner models and advanced heterogeneous ensemble models. Although it cannot be definitely stated that any of these models significantly outperformed others, the results highlight the feasibility and strong predictive capabilities of the approaches employed. Future research will focus on utilizing larger datasets and exploring more advanced AI models to further enhance prediction accuracy and robustness.

## Figures and Tables

**Figure 1 sensors-25-04402-f001:**
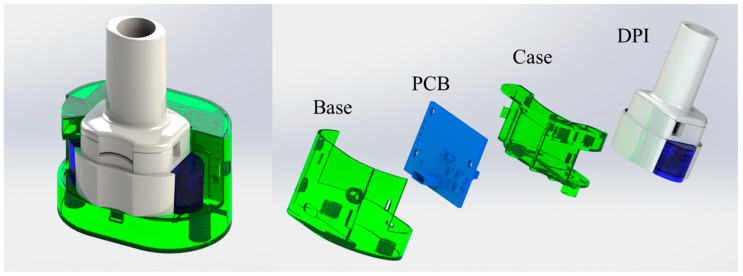
Trimetric view (**left**) and exploded view (**right**) of the custom-designed DPI digital monitoring system.

**Figure 2 sensors-25-04402-f002:**
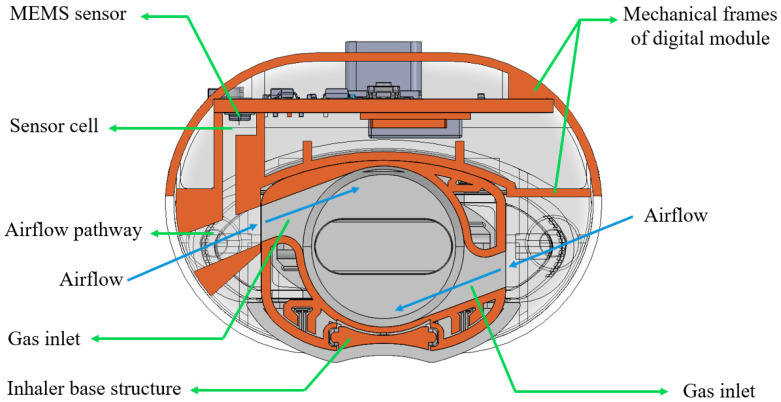
The cross-sectional view of the customized digital module with the dry powder inhaler.

**Figure 3 sensors-25-04402-f003:**
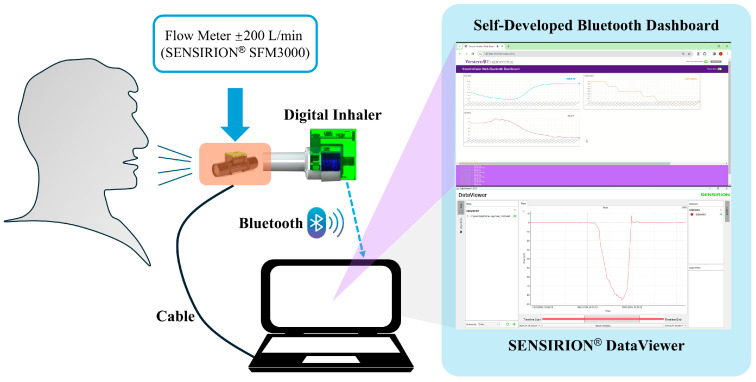
Experimental setup for the collection of inhalation data.

**Figure 4 sensors-25-04402-f004:**
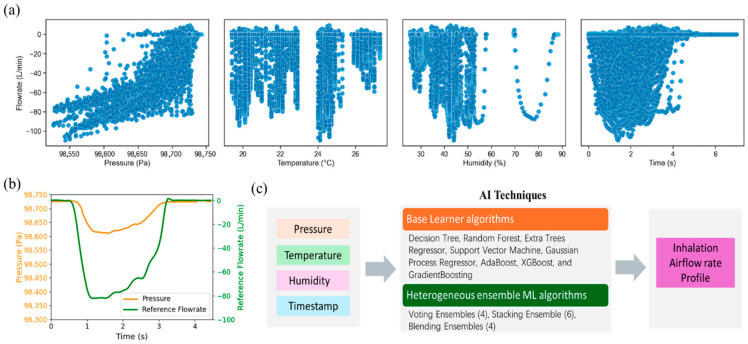
Illustration of collected raw data. (**a**) scatter plot of collected signals vs. reference flowrate; (**b**) case sample of collected pressure vs. time and referenced flowrate vs. time profiles; (**c**) flow chart of AI techniques for real-time prediction of inhalation airflow rate.

**Figure 5 sensors-25-04402-f005:**
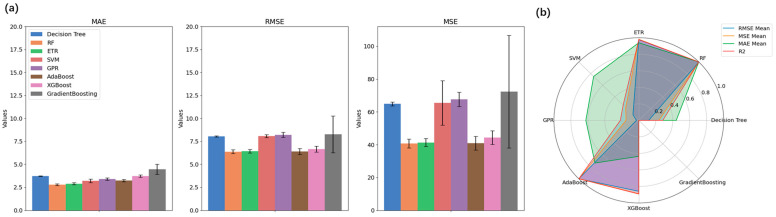
(**a**) Bar plots of errors of different algorithms for cross-validation results; (**b**) Radar plot for model selection.

**Figure 6 sensors-25-04402-f006:**
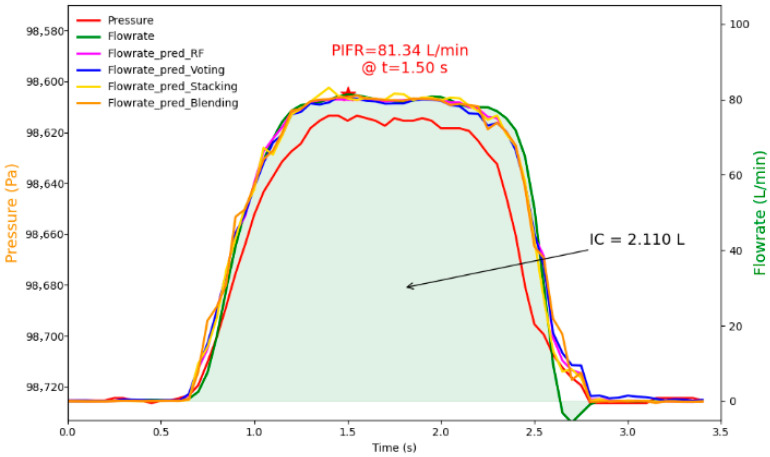
Illustration of collected pressure data profile, referenced flowrate profile, and predicted flowrate profiles using the best base learner, voting ensemble, stacking ensemble, and blending ensemble models.

**Table 1 sensors-25-04402-t001:** Machine learning model performance for different models on training, cross-validation, and testing subsets.

	Training				Cross-Validation				Testing			
ML Algorithms	R^2^	RMSE	MSE	MAE	R^2^	RMSE	MSE	MAE	R^2^	RMSE	MSE	MAE
**Decision Tree**	0.926	7.345	53.945	3.459	0.911	8.053	64.911	3.720	0.919	7.541	56.871	3.383
**Random Forest (RF)**	0.983	3.547	12.583	1.507	**0.944**	**6.383**	**40.781**	**2.788**	0.952	5.833	34.026	**2.448**
**Extra Trees Regressor (ETR)**	0.975	4.237	17.949	1.886	0.943	6.428	41.359	2.889	0.951	5.846	34.174	2.556
**Support Vector Machine (SVM)**	0.914	7.914	62.639	3.049	0.910	8.089	65.502	3.201	0.917	7.629	58.207	2.888
**Gaussian Progress Regressor**	**0.986**	**3.175**	**10.083**	**1.253**	0.908	8.221	67.630	3.415	0.929	7.081	50.142	2.893
**AdaBoost**	0.979	3.941	15.529	2.403	**0.944**	6.386	40.890	3.238	**0.955**	**5.635**	**31.759**	2.880
**XGBoost**	0.973	4.430	19.625	2.587	0.939	6.651	44.335	3.730	0.948	6.029	36.343	3.404
**GradientBoosting**	0.917	7.811	61.012	4.092	0.898	8.273	72.401	4.456	0.924	7.297	53.242	3.803

Note: the best performance is marked in bold.

**Table 2 sensors-25-04402-t002:** Evaluation results of heterogeneous ensemble models on cross-validation sets.

	R^2^		RMSE		MSE		MAE	
ML Algorithms	Mean	SD	Mean	SD	Mean	SD	Mean	SD
**Random Forest (RF)**	0.944	0.003	6.383	0.210	40.781	2.661	2.788	0.110
**AdaBoost**	0.944	0.004	6.386	0.322	40.890	4.212	3.238	0.117
**Extra Trees Regressor**	0.943	0.003	6.428	0.192	41.359	2.460	2.889	0.113
**XGBoost**	0.939	0.005	6.651	0.320	44.335	4.200	3.730	0.130
**Voting Ensemble_1**	0.947	0.004	6.194	0.252	38.433	3.082	3.100	0.130
**Voting Ensemble_2**	0.947	0.003	6.233	0.230	38.899	2.835	3.093	0.124
**Voting Ensemble_3**	0.945	0.003	6.352	0.213	40.385	2.680	2.813	0.111
**Voting Ensemble_4**	0.946	0.003	6.271	0.237	39.385	2.959	3.233	0.123
**Stacking Ensemble_1**	**0.948**	0.003	6.184	0.233	38.295	2.864	2.806	0.118
**Stacking Ensemble_2**	0.933	0.002	7.017	0.170	49.268	2.395	3.605	0.151
**Stacking Ensemble_3**	**0.948**	0.004	6.174	0.264	38.193	3.246	2.797	0.130
**Stacking Ensemble_4**	0.941	0.003	6.588	0.253	43.467	3.350	2.950	0.108
**Stacking Ensemble_5**	0.930	0.002	7.150	0.154	51.144	2.196	3.334	0.114
**Stacking Ensemble_6**	0.938	0.004	6.699	0.236	44.939	3.144	3.095	0.138

Note: the best performance is marked in bold.

**Table 3 sensors-25-04402-t003:** Evaluation performance of machine learning models on testing sets.

ML Algorithms	R^2^	RMSE	MSE	MAE
**Decision Tree**	0.919	7.541	56.871	3.383
**Random Forest (RF)**	0.952	5.833	34.026	2.448
**Extra Tree Regressor (ETR)**	0.951	5.846	34.174	2.556
**Support Vector Machine (SVM)**	0.917	7.629	58.207	2.888
**Gaussian Progress Regressor**	0.929	7.081	50.142	2.893
**AdaBoost**	0.955	5.635	31.759	2.880
**XGBoost**	0.948	6.029	36.343	3.404
**GradientBoosting**	0.924	7.297	53.242	3.803
**Voting Ensemble_1**	0.956	5.570	31.029	2.750
**Voting Ensemble_2**	0.956	5.590	31.252	2.746
**Voting Ensemble_3**	0.952	5.784	33.456	2.471
**Voting Ensemble_4**	0.955	5.603	31.392	2.887
**Stacking Ensemble_1**	0.956	5.565	30.972	2.449
**Stacking Ensemble_2**	0.942	6.406	41.033	3.450
**Stacking Ensemble_3**	0.957	5.456	29.769	2.403
**Stacking Ensemble_4**	0.953	5.771	33.309	2.500
**Stacking Ensemble_5**	0.937	6.679	44.615	3.013
**Stacking Ensemble_6**	0.944	6.249	39.051	2.952
**Blending Ensemble_1**	0.952	5.828	33.967	2.593
**Blending Ensemble_2**	0.936	6.722	45.181	3.131
**Blending Ensemble_3**	0.953	5.754	33.103	2.553
**Blending Ensemble_4**	0.938	6.590	43.431	2.920

**Table 4 sensors-25-04402-t004:** The estimation accuracy of PIFR and IC across the whole datasets using the best base learner, voting ensemble, stacking ensemble, and blending ensemble models.

ML Models	PIFR (%)	IC (%)
**Best Single Base Learner (Random Forest)**	97.7 ± 2.9	95.2 ± 9.0
**Best Voting Ensemble (Voting Ensemble 1)**	96.5 ± 4.5 *	94.3 ± 9.5 *
**Best Stacking Ensemble (Stacking Ensemble 3)**	96.7 ± 3.9 *	95.1 ± 9.2
**Best Blending Ensemble (Blending Ensemble 3)**	95.7 ± 5.1 *	94.3 ± 9.5 *

* represents statistically significant with *p* < 0.05, compared with the Random Forest model.

## Data Availability

The raw data supporting the conclusions of this article will be made available by the authors on request.

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
