# Peer review of "Development of an AI-Empowered Novel Digital Monitoring System for Inhalation Flow Profilesâ€"

_sensors, 2025, doi:10.3390/s25144402_

Round 1
Reviewer 1 Report
Comments and Suggestions for Authors
This paper presents a digital monitoring system that monitors pressure during dry powder inhaler (DPI) use. This system integrates sensor data with machine learning techniques to estimate inhalation flow profiles based on the collected data. The results show that inhalation flow profiles can be estimated with an accuracy of 97.7% for peak inspiratory flow rate (PIFR) and 95.2% for inspiratory capacity (IC). The paper addresses an important and practical challenge in healthcare. However, to enhance the manuscript's quality and impact, the authors should address the following points:
1. Although the authors often use terms such as "developed machine learning models", it should be clarified that this paper does not actually develop a new machine learning algorithm. Instead, the study relies on existing, well-established ML methods (e.g., Random Forest, Decision Trees, SVM, GPR, XGBoost, etc.) which are trained, fine-tuned (using grid search), and comparatively evaluated for the specific application of predicting inhalation flow profiles. This is a valid and relevant approach. However, the authors may consider reformulating to better reflect the true scope of their contribution, such as stating that "existing ML algorithms were selected, applied and optimized for the problem of inhalation flow estimation."
2. The manuscript reports high prediction accuracy, but the implementation details are not worked out. Authors should include more information about computational requirements, training time, libraries used, ...
3. There are some grammatical errors and typos throughout the manuscript. A thorough proofreading is necessary to enhance readability and ensure the professionalism of the presentation. (e.g., "Following by the data collection" - "Following data collection")
(In this section, the top-performed base leaner model (RF) - .... learner)
The repetition of the text in the following part of the abstract is noticed:
“As a possible strategy, digital monitoring system coupled with dry powder inhalers (DPIs) has emerged to estimate flow profiles and provide inhalation information. As well, estimation has also been improved with the use of artificial intelligence (AI). A potential strategy involves the integration of digital monitoring system with dry powder inhalers (DPIs) to estimate the flow profiles and provide feedback to patients, with further improvements facilitated by the application of artificial intelligence (AI).”
It is recommended that the authors correct the text in order to avoid repeating both the text and abbreviations.
4. Although the mentioned techniques for evaluating the performance of the model are generally known, the authors state the equations y of them, but they should also define the parameters used in the equations (e.g. y).
5. In Figure 4, the parts of the diagram under items a), b) and c) are not clearly marked.
The font of the letters on the diagrams in Figure 5 should be increased to make them readable, because now it is impossible to read what is written on the axes and in the title of the diagram.
6. The reference list must be corrected, e.g. reference 3, ref. 30, ref. 32, ref. 39, ref. 43.
Comments on the Quality of English LanguageA thorough proofreading is necessary to enhance readability.
Reviewer 2 Report
Comments and Suggestions for Authors
This paper describes a novel device for monitoring the use of dry powder inhalers. Misuse of DPIs may indeed decrease the therapy effectiveness. Therefore, such a new device could be beneficial for patients initiating DPI therapy.
Abstract - the first sentence is not acceptable. Perhaps better to write that the use of DPIs is a cornerstone of the therapy of chronic pulmonary diseases.
In conclusion: 'to the estimation' - better 'to assess'
Introduction: Could be shorter.
Results: Please consider presenting some of them as a supplementary file.
Discussion:
It is not true that authors proved the cost-effectiveness of a new digital monitoring system. In fact, the obtained results showed only high accuracy for the estimation of inhalation flow parameters, which could be further tested in patients with chronic pulmonary diseases.
Consequently, the concussion section needs to be corrected.
The introduction could be better
Reviewer 3 Report
Comments and Suggestions for Authors
This manuscript presents a novel, cost-effective digital monitoring system for dry powder inhalers (DPIs), which integrates MEMS sensors and advanced machine learning algorithms to accurately estimate inhalation flow profiles. The study is well-structured, and the methodology is rigorous, combining hardware development and AI-based data analysis. The results are promising, with high prediction accuracy for key inhalation parameters such as peak inspiratory flow rate (PIFR) and inspiratory capacity (IC). Overall, the topic is timely and relevant, and the manuscript has strong potential for publication in Sensors.
However, there are several important issues that need to be addressed prior to acceptance.
- Sample Size and Generalizability
Issue: The study includes only eight healthy volunteers, which limits the diversity and generalizability of the dataset. No data from patients (e.g., with asthma or COPD) are included.
Recommendation: Please discuss the limitation of using a small and homogeneous dataset and emphasize the need for future studies involving clinical patients to validate model robustness in real-world scenarios.
- Clinical Relevance of Prediction Accuracy
Issue: Although the model achieves high accuracy in estimating PIFR and IC, it is unclear how these values translate to clinical decision-making. For example, how do small prediction errors impact the determination of sufficient inhalation effort?
Recommendation: Consider discussing whether the model’s accuracy is sufficient for detecting clinically relevant thresholds, and clarify any acceptable error margins in this context. A brief binary classification analysis (achieving vs. not achieving target PIFR) could be illustrative.
- Deployability of Ensemble Models
Issue: The system utilizes complex ensemble models (e.g., Stacking Ensemble), but no discussion is provided regarding their feasibility for real-time deployment on embedded devices.
Recommendation: Please include a brief discussion on model deployment feasibility, particularly with respect to computational efficiency, latency, and compatibility with low-power hardware.
- Statistical Significance of Model Comparison
Issue: While multiple models are compared in terms of R² and error metrics, the statistical significance of these differences is not addressed.
Recommendation: Clarify whether the performance differences among the models are statistically significant. If not analyzed quantitatively, please acknowledge this as a limitation.
- Figure Presentation
Issue: Some figures (Figure 8) lack complete labeling, such as units or legends, which may affect clarity.
Recommendation: Ensure that all figures include axis labels, units, and legends where necessary to improve readability and standalone interpretability.
Reviewer 4 Report
Comments and Suggestions for Authors
The manuscript presents original and sufficient content, meeting the publication’s criteria. Both the subject matter and the formal tone are well-suited to the journal’s audience. The writing style and clarity are appropriate. Although the article provides extensive information, I recommend that the authors revise the paper.
- The abstract currently repeats similar ideas about the integration of digital monitoring systems with dry powder inhalers (DPIs) and the use of artificial intelligence (AI) for estimation.
- In Abstract the authors must replace phrases like "As a possible strategy" and "A potential strategy involves" with more direct language.
- The literature review mostly summarizes existing technologies but could include more critical analysis—such as comparing the performance, scalability, and real-world adoption of different systems.
- Although limitations of previous methods are mentioned, the review could more explicitly identify the specific research gap that the current work addresses, making the motivation for the study clearer.
- While the review cites a range of studies, it could benefit from more recent references, especially on the latest developments in AI-based inhaler monitoring and commercial device validation.
- The presentation of quantitative data must be more clearly: authors must report sample sizes (number of tests, patients, or data points).
- Authors must provide summary statistics (mean, standard deviation, confidence intervals) alongside accuracy percentages.
The abstract currently repeats similar ideas about the integration of digital monitoring systems with dry powder inhalers (DPIs) and the use of artificial intelligence (AI) for estimation.
Round 2
Reviewer 2 Report
Comments and Suggestions for Authors
Thank you for the collaboration—no further comments.
Reviewer 4 Report
Comments and Suggestions for Authors
I agree with the revised version of the manuscript.